# Technical note: Measurements of fluorescent dissolved organic matter (FDOM) in seawater (Filter blanks, pore sizes, and storage)

Junhyeong Seo[1,2*], Heejun Han[1,2], Intae Kim[2], and Guebuem Kim[1]

[1]School of Earth and Environmental Sciences/Research Institute of Oceanography, Seoul National University, Seoul 08826, South Korea
[2]Marine Environmental Research Department, Korea Institute of Ocean Science and Technology (KIOST), Busan 49111, South Korea

*Correspondence to*: Junhyeong Seo (junhyeong@kiost.ac.kr)

**Abstract.**

Fluorescent dissolved organic matter (FDOM) provides crucial information regarding the sources and characteristics of dissolved organic matter (DOM) in oceans. However, results from FDOM measurement can depend on filter blanks, pore sizes, and sample storage. To develop more reliable methods for FDOM measurements, we examined uncertainties associated with different preparation methods for seawater samples. Three primary components were identified from these samples using parallel factor analysis: terrestrial humic-like peak (C peak), marine humic-like peak (M peak), and protein-like peak (T peak). Relatively high blank values were observed when samples were filtered through a pre-combusted glass fiber filter (Whatman, borosilicate, 0.7 μm, 47 mm) and a membrane filter (Whatman, mixed cellulose ester, 0.2 μm, 47 mm) without pre-cleaning. These blank values were negligible when both filters were washed with 5 mL of 0.1 M HCl (~ 0.29 mL cm$^{-2}$) or 20 mL of distilled water (~1.16 mL cm$^{-2}$). The effects of different filter pore sizes were not observed for the C and M peaks, but lower T peak values were observed for filtered samples relative to unfiltered samples. During storage, C and M peaks showed consistent results for 21 days (8% ± 3%) when kept in pre-combusted amber glass vials in a refrigerator or a freezer. In contrast, clear changes were observed in samples stored at room temperature after five days. Thus, reliable C and M peaks can be obtained from unfiltered or filtered samples stored in a refrigerator or freezer for up to three weeks. However, T peak concentration decreased rapidly in both filtered (15%–50%) and unfiltered samples (10%–40%) within five days, indicating the influence of significant biological and abiotic processes. Therefore, our results suggest that careful sample filtration, storage, and blank controls are necessary for T peak measurements.

## 1. Introduction

Fluorescent dissolved organic matter (FDOM), which emits fluorescent light after absorbing energy, is ubiquitous in the ocean and provides important information on the origins and behavior of dissolved organic matter (DOM) in the ocean (Nelson and Siegel, 2013; Stedmon and Nelson, 2015). FDOM in ocean waters is generally classified into two groups (humic-like and protein-like substances) based on the excitation and emission spectrum (Coble, 2007). The humic-like component is primarily derived from microbial decomposition of organic debris in sediments and soils and from materials sinking through the water column as marine snow (Yamashita and Tanoue, 2008; Yamashita and Tanoue, 2009). In contrast, the protein-like component represents a more labile fraction of DOM and is mainly produced through primary production and biological activity in the surface waters (Lønborg et al., 2010). Based on distinct fluorescent properties, FDOM has been used as a tracer for water circulation in the ocean (GonçAlves-Araujo et al., 2016; Galletti et al., 2019; Margolin et al., 2018), estimating DOM turnover times in the global ocean (Catalá et al., 2015), and for calculating the fractions of different water masses in the ocean (Kim et al., 2020; Wang et al., 2022).

Over the last few decades, the measurements of FDOM in the ocean have been extensively conducted. Accordingly, various sampling and laboratory protocols have been developed in different laboratories without intercalibrations in sampling, storage, and measurements. In general, freezing and refrigeration have been used to store samples. The freezing of seawater samples after filtering with a pre-combusted (4 h, 450°C) glass fiber filter (GF/F, Whatman, 0.7 μm pore-size, 47 mm diameter) is one of the widely used methods to preserve samples when measurements are delayed (> one month) (Conmy et al., 2009; Yamashita et al., 2021). However, FDOM concentration can vary up to ± 50% during the freezing and thawing process due to aggregation and disaggregation, especially when high levels of humic substances are present (Murphy et al., 2013b; Spencer et al., 2007). Thus, FDOM sampling was commonly performed by filtering the water sample (~ 40 mL) with pre-combusted (4 h, 450°C) GF/F and storing it in a pre-combusted (4 h, 450°C) amber vial without any treatment. In some studies, different types of membrane filters, such as cellulose acetate, polyethersulfone, and polycarbonate, have also been used for FDOM measurement (Amaral et al., 2023; Chen et al., 2022; Vines and Terry, 2020). Then, the FDOM samples were generally kept in a refrigerator (4°C) before the measurement (Coble et al., 1998; Stedmon et al., 2003). However, the potential uncertainties that can be introduced by these various sampling and storage methods, particularly in different marine environments, have not

been evaluated carefully yet. Therefore, in this study, we measured FDOM concentrations in open- and coastal-ocean samples under different conditions (i.e., filter pore sizes and storage strategies) to obtain reliable methods for FDOM measurements.

## 2. Methods

### 2.1. Procedural blank

We used pre-combusted GF/F and membrane filters (Whatman, 0.2 μm pore-size, 47 mm diameter) to examine the filter blanks of FDOM measurements. The materials of GF/F and membrane filter are composed of borosilicate and mixed cellulose ester, respectively. Hereafter, membrane filter refers specifically to the mixed cellulose ester filter described above. The filter blank was tested with a sequential filtration process, adding up to 100 mL of distilled water and 0.1 M HCl, respectively. The measurement of FDOM was conducted by collecting 5 mL water samples at the volume of 5, 10, 15, 20, 30,

50, and 100 mL during the sequential filtration process.

### 2.2. Sampling and preparation

  Sampling was conducted in the East Sea (Japan Sea) (Station EC1; 37.33°N, 131.45°E) in April 2019 and in the Jinhae Bay (JH; 35.04°N, 128.62°E) in August 2019. The water samples (3 m, 300 m, and 2300 m depths) of Station EC1 were collected using a Niskin sampler onboard a ship (*R/V* Ieordo). The surface water samples (~0.5 m) from the JH were

collected at three sites (JH1, JH2, and JH3). The samples from the JH were collected using a pump system onboard a ship. All water samples were stored in pre-cleaned 20 L polypropylene bottles in a refrigerator and transported to the land-based laboratory without any treatment.

  In the laboratory, filtration was performed using a 47 mm diameter funnel system that had been cleaned with 1 M HCl. The funnel and filter system were rinsed with distilled water and seawater samples. Unfiltered and filtered (0.7 or 0.2

μm) water samples (~ 40 mL each) were stored in the dark (i.e., in pre-combusted amber vials) to prevent photodegradation and kept at three different temperatures (–20℃, 4℃, or 25℃). All the samples were triplicated. Thus, each sample was divided into 27 amber vial samples. The initial measurement was conducted within two days after seawater sampling. The measurement interval to examine the effect of storage time was 1, 3, 5, 7, 14, and 21 days, respectively, from the initial measurement.

## 2.3. Analytical protocols

FDOM fluorescence intensity was determined by a spectrophotometer (Aqualog, Horiba, USA). We used 10 mm path-length quartz cuvettes, which went through the signal-to-noise test. The 10 mm path-length quartz cuvettes have been widely used to measure FDOM in the marine environment (Murphy et al., 2013a). The measurement of ultra-pure water (milli-Q water, < 18.2 Ω) was performed at the start of FDOM analysis, and the result was considered as a blank value. The excitation-emission matrices (EEMs) were collected over excitation wavelengths ranging from 240 to 700 nm at 3 nm intervals, and

emission wavelengths ranging from 250 to 500 nm at 5 nm intervals. The integration time of EEMs was 5 seconds. The parallel factor analysis (PARAFAC) model was utilized through Solo software (Eigenvector Inc., USA) to identify and characterize key fluorescent components (Han et al., 2022). Before applying the model, corrections for the inner-filter effect were conducted to reduce distortions in fluorescence measurements (Kothawala et al., 2013). To ensure the reliability of the extracted components, the model was assessed through split-half validation and random initialization, confirming its robustness (Bro,

1997; Stedmon and Bro, 2008; Zepp et al., 2004). Since fluorescence intensity is highly instrument-dependent, the intensity of FDOM was normalized by the Raman peak area using ultra-pure water to convert to Raman Unit (R.U.). The R.U. value represented the integrated area of the water Raman peak at an excitation wavelength of 350 nm (Lawaetz and Stedmon, 2009).

To compare the intensity of all samples, the PARAFAC model was applied to a single data set of the EEM data, including filter blank samples (distilled water and 0.1 M HCl) as well as open- and coastal-ocean samples, while considering

filter pore sizes, storage time, and temperature for the seawater samples. Terrestrial humic-like peak (C peak, Ex/Em = 375/457 nm), marine humic-like peak (M peak, Ex/Em = 315/391 nm), and protein-like peak (T peak, Ex/Em = 270/313 nm) were identified by the PARAFAC model. These peak positions were assigned based on the previously reported values (Coble, 1996; Coble et al., 1998; Coble, 2007). The fluorescence spectra results were compared with the OpenFluor spectra database (Murphy et al., 2014), and statistical matches were found at a confidence level of > 95%, with 28 matches for the C peak, 37 for the M

peak, and 45 for the T peak. The EEM contours and loading results from the PARAFAC model are presented in Figure 1.

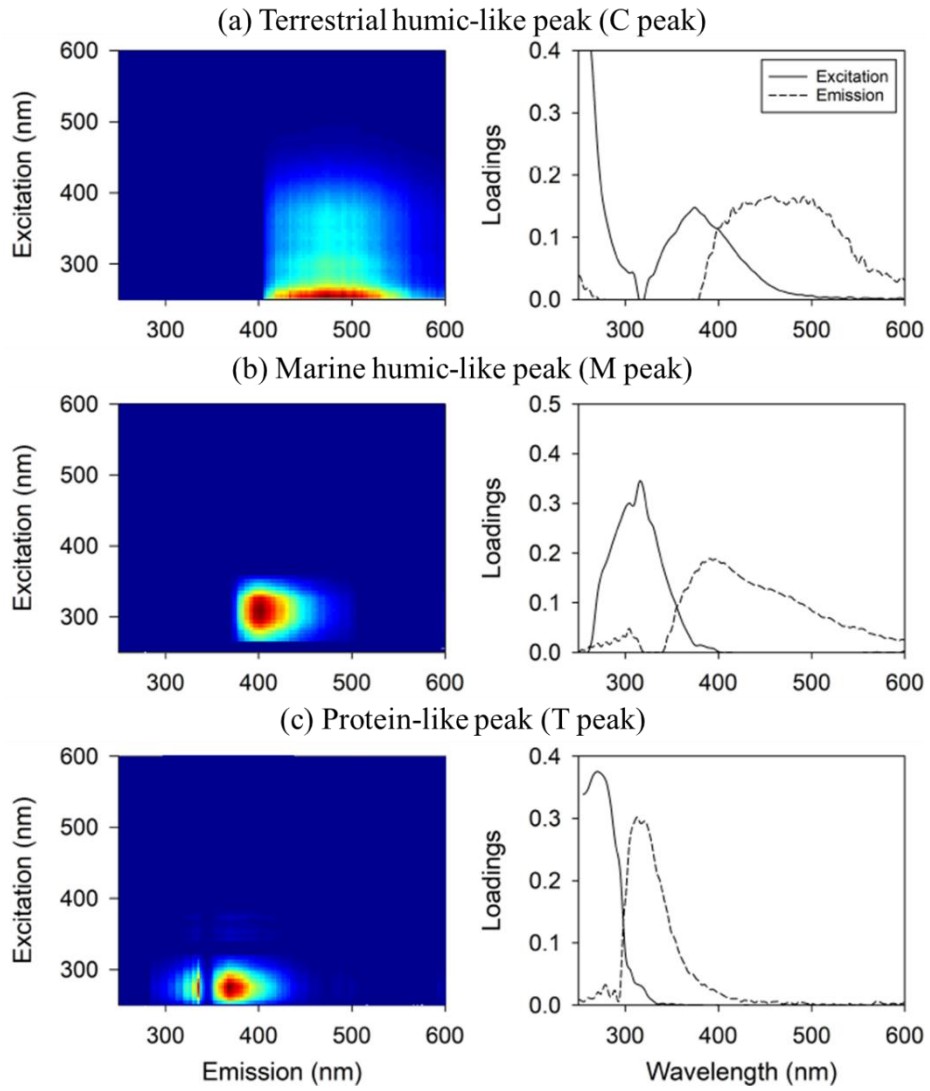

**Figure 1: EEM contours and loading results from the PARAFAC model include: (a) terrestrial humic-like peak, (b) marine humic-like peak, and (c) protein-like peak.**

## 3. Results

### 3.1. Filter blanks

During GF/F filtration, the concentrations of the C and T peaks measured in the filtrate were negligible for both distilled water-washed and acid-treated filters (Fig. 2). Unlike the C and T peaks, a significantly high concentration of the M peak (up to 0.15 R.U.) was observed in the filtrate when the filter was washed less than 20 mL of distilled water (volume per

filter surface area: ~1.16 mL cm$^{-2}$), corresponding to up to 60% of the M peak in the coastal ocean (Station JH) samples. The

acid-washed filter showed a negligible concentration of the M peak.

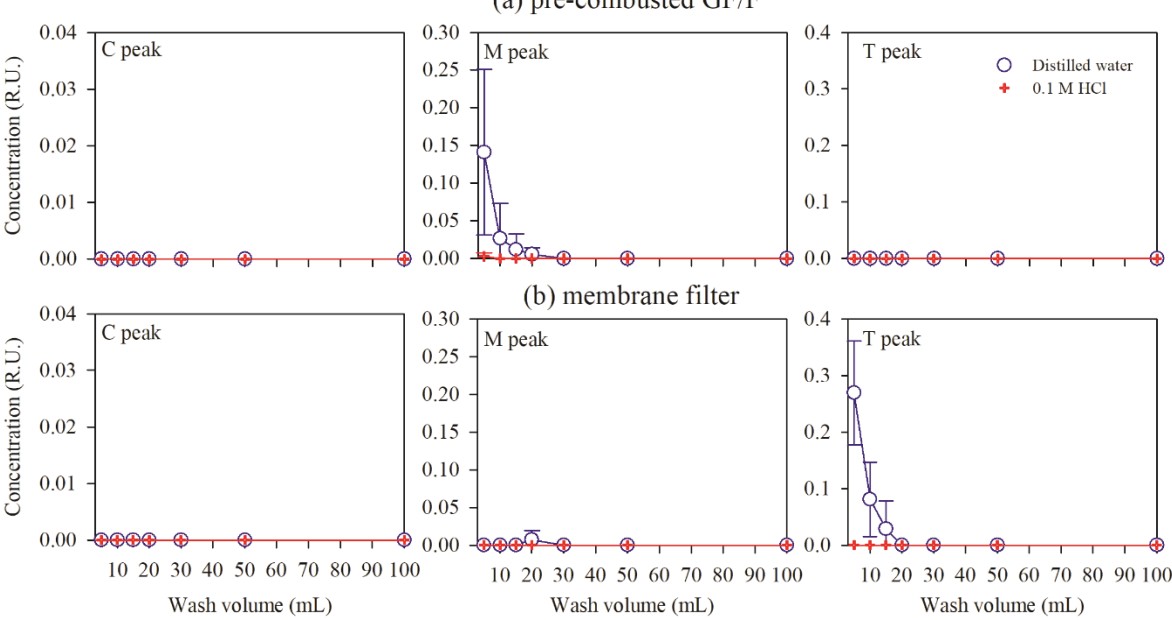

**Figure 2: Filter blanks of FDOM concentrations for (a) pre-combusted GF/F and (b) membrane filters washing with distilled water or 0.1 M HCl.**

During filtration with the membrane filter, the concentrations of C and M peaks measured on the filtrate were

negligible for both washing with distilled water and acid-treated filters (Fig. 2). However, the T peak showed high

concentrations in the filtrate of which the filter was washed with distilled water, particularly before 20 ml of filtration (up to

0.27 R.U.). The blank of the T peak was almost 95% in the open ocean (Station EC1) samples. The acid-washed filter exhibited

a negligible filter blank for the T peak.

**3.2. Filter pore sizes**

The initial concentrations of the C, M, and T peaks at 3 m depth at station EC1 from unfiltered samples were 0.44,

0.78, and 0.35 R.U., respectively. At 300 m, the concentrations of the C, M, and T peaks were 0.53, 0.97, and 0.26 R.U., and

at 2300 m, they were 0.63, 0.64, and 0.21 R.U. (Table 1). The concentrations of the C and T peaks in the open ocean showed

no clear differences across depths, whereas the concentration of the M peak was relatively higher at 300 m compared to the

other depths. In the coastal ocean, at stations JH1, JH2, and JH3, the concentrations of the C, M, and T peaks were 0.89, 0.24, and 0.88 R.U. at JH1; 0.98, 0.25, and 1.08 R.U. at JH2; 1.01, 0.27, and 1.59 R.U. at JH3. For samples filtered through 0.7 μm GF/F and 0.2 μm membrane filters, the initial concentrations of the C and M peaks at 3, 300, and 2300 m in the open ocean, and coastal stations JH1, JH2, and JH3, are similar to those observed from unfiltered samples and also summarized in Table 1. For the T peak, the concentrations in samples filtered through the 0.7 μm GF/F were 0.28, 0.23, and 0.18 R.U. at 3, 300, and

2300 m in the open ocean, respectively, and 0.43, 0.43, and 0.53 R.U. at coastal stations JH1, JH2, and JH3, respectively. Similar T peak concentrations were observed in samples filtered through the 0.2 μm membrane filter.

**Table 1. Initial concentrations of C, M, and T peaks in unfiltered and filtered samples from the East Sea and Jinhae Bay.**

| Station | Depth (m) | C peak (R.U.) | | | M peak (R.U.) | | | T peak (R.U.) | | |
|---|---|---|---|---|---|---|---|---|---|---|
| | | Unfiltered | 0.7 μm | 0.2 μm | Unfiltered | 0.7 μm | 0.2 μm | Unfiltered | 0.7 μm | 0.2 μm |
| | 3 | 0.44 | 0.44 | 0.44 | 0.78 | 0.79 | 0.77 | 0.35 | 0.28 | 0.28 |
| EC1 | 300 | 0.53 | 0.54 | 0.47 | 0.97 | 0.95 | 0.89 | 0.26 | 0.23 | 0.18 |
| | 2300 | 0.63 | 0.66 | 0.67 | 0.64 | 0.63 | 0.63 | 0.21 | 0.18 | 0.19 |
| JH1 | surface | 0.89 | 0.90 | 0.82 | 0.24 | 0.21 | 0.21 | 0.88 | 0.43 | 0.39 |
| JH2 | surface | 0.98 | 0.94 | 0.94 | 0.25 | 0.22 | 0.21 | 1.08 | 0.43 | 0.37 |
| JH3 | surface | 1.01 | 0.97 | 1.00 | 0.27 | 0.24 | 0.25 | 1.59 | 0.53 | 0.50 |

The concentrations of C and M peaks for unfiltered samples were similar to those for the filtered samples (0.7 μm)

in the open ocean (97% ± 1%, t-test, $p > 0.05$) and coastal ocean (96% ± 4%, $p > 0.05$) (Fig. 3). The $p$-value higher than 0.05 indicated that the differences between the unfiltered and filtered samples were not statistically significant. The concentrations of T peak in unfiltered samples from the open ocean showed a slight difference with those in filtered samples (0.7 μm) (83% ± 9%, $p = 0.05$). Unlike the open ocean, the concentrations of T peak in unfiltered samples from the coastal ocean were 48%– 79% higher than those observed in filtered samples (0.7 μm or 0.2 μm), and this difference was statistically significant ($p <$

0.05). In both open- and coastal-ocean samples, no significant difference in FDOM concentrations was observed between the 0.7 μm and 0.2 μm filtration.

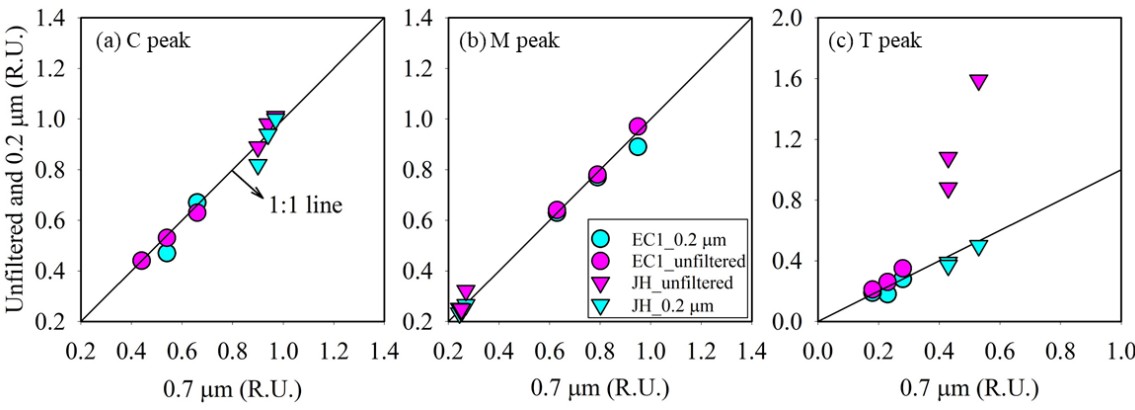

**Figure 3: Initial concentrations of (a) C peak, (b) M peak, and (c) T peak in samples obtained from the open ocean and the coastal ocean, depending on the filter pore sizes. Circles and triangles indicate the samples from the East Sea and Jinhae Bay, respectively. The solid line indicates the 1:1 line.**

### 3.3. Storage strategies

The concentrations of C and M peaks for unfiltered and filtered samples (0.7 and 0.2 μm) from the open- and coastal-ocean stored in the refrigerator (4°C) and freezer (–20°C) under dark conditions, showed no clear differences between the initial and after 21 days measurements (8% ± 3%, $p > 0.05$) (Fig. 4). At room temperature, the concentrations of C and M peaks in the unfiltered or filtered samples from the open- and costal-ocean also showed no clear differences within five days (7% ± 2%, $p > 0.05$) (Fig. 5).

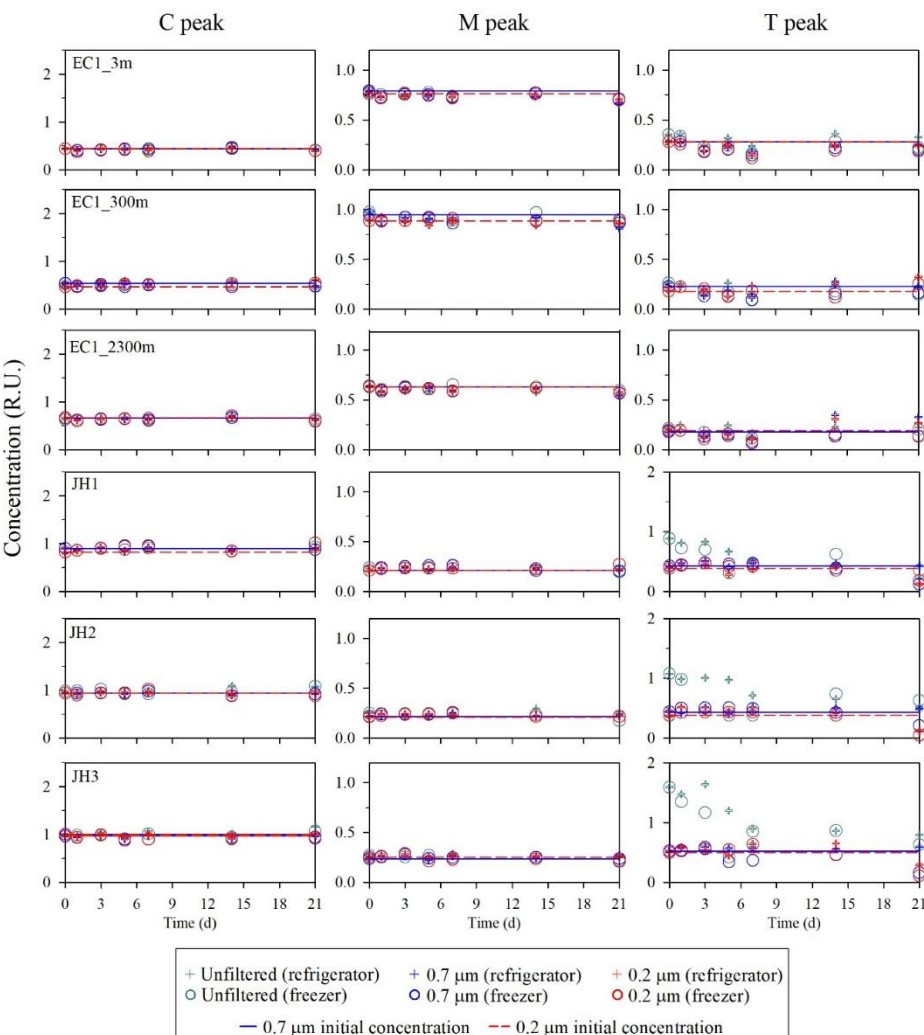

**Figure 4: Changes in FDOM concentrations after being stored in the refrigerator or freezer for samples obtained from the open and coastal oceans. Circles and crosses represent the refrigerator and freezer, respectively. Solid and dashed lines denote the initial concentration of FDOM after filtration with 0.7 µm pore size (blue) and 0.2 µm pore size (red) filters, respectively.**

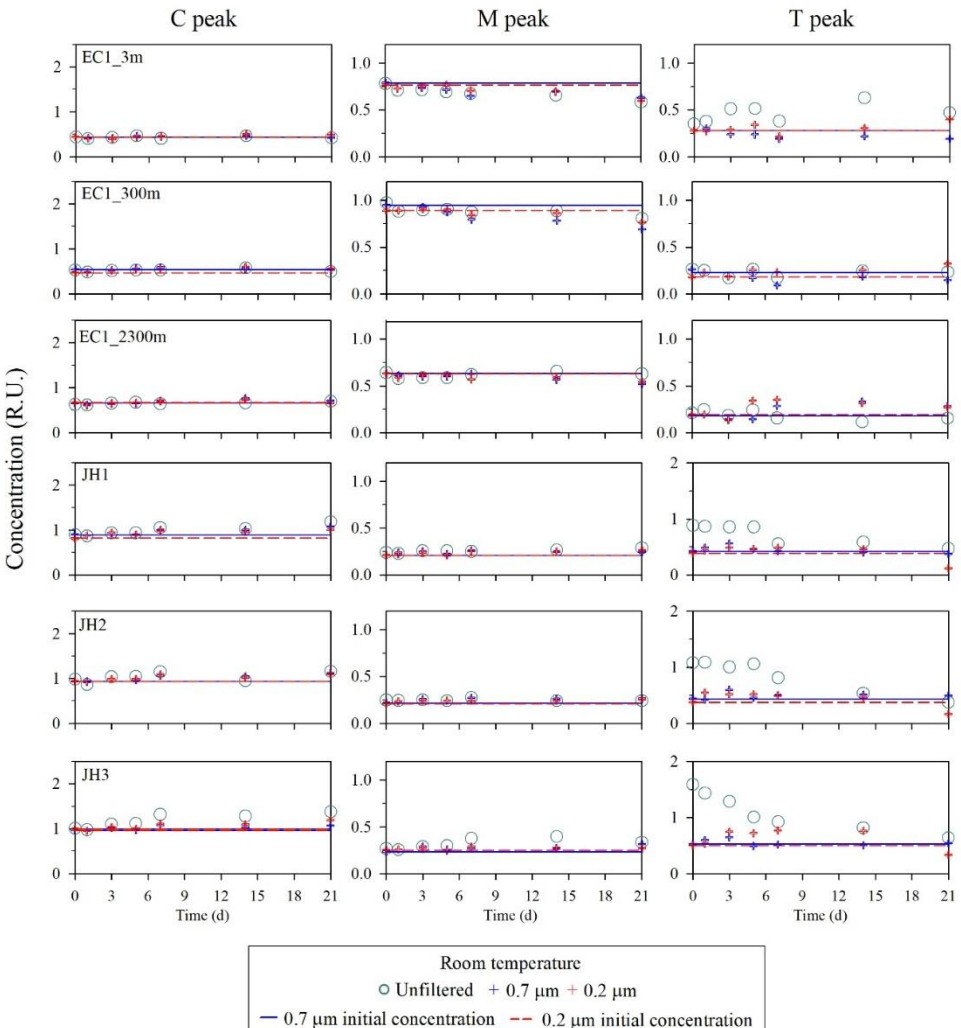

**Figure 5: Changes in FDOM concentrations after being stored at room temperature for samples in the open and coastal oceans. Circles and crosses represent the unfiltered and filtered samples, respectively. Solid and dashed lines denote the initial concentration of FDOM after filtering by 0.7 μm pore size (blue) and 0.2 μm pore size (red) filters, respectively.**

However, regardless of the open or coastal ocean, the concentrations of T peak in unfiltered or filtered samples showed a variation after five days compared with the initial value ($24\% \pm 5\%$, $p < 0.05$). The difference of T peak increased more significantly after 21 days in unfiltered ($42\% \pm 3\%$, $p < 0.05$) and filtered samples ($43\% \pm 15\%$, $p < 0.05$).

## 4. Discussion

The accuracy of FDOM measurements can be largely affected by the filtration process. Pre-combusted GF/F has been widely used for water filtration for FDOM sampling due to its advantages in low DOM backgrounds after ignition, high flow rate, and large capacity. However, we observed high contamination of the M peak even after washing with distilled water. This seems to be produced by filter fiber particles from the ashed filter. This result suggests that GF/F should be washed with 20 mL of distilled water (~1.16 mL cm$^{-2}$) or 5 mL of 0.1 M HCl (~0.29 mL cm$^{-2}$) before the sample filtration for FDOM measurements. Although GF/F has such advantages, the large filter pore size (0.7 µm) can allow the passage of microorganisms or colloids (Tanoue, 1992), which could mislead the measurement due to the biological activity and abiotic activity, such as cell bursting and absorption/desorption on colloidal particles.

To prevent these effects, a membrane filter with a 0.2 µm pore size has been often used for FDOM sampling (Rochelle-Newall and Fisher, 2002). However, we observed high blank values of the membrane filter for the T peak without pre-washing using 20 mL of distilled water or 5 mL of 0.1 M HCl. Our results from the blank test for different filters display that the filter blank may introduce uncertainties in the measurement of FDOM in seawater. This is particularly noticeable for the M peak in open ocean samples and the T peak in the coastal ocean samples. Therefore, careful pre-washing, including ashing processes, is necessary to prevent any contamination from filtration. In cases where distilled water or acid is not available in the field, pre-washing the filter with sample water may be a practical alternative to reduce potential contamination before filtration.

The concentrations of all FDOM components in the open ocean showed no significant differences between unfiltered and filtered samples (0.7 and 0.2 µm). This result suggested that the presence of particles in the sample did not significantly affect the fluorescent concentration in the open ocean. These results also indicated that filtration may not be essential for measuring C and M peaks in open ocean waters, and measurements can be reliably conducted without filtration. Similarly, filtration was not conducted in the open-ocean samples to avoid potential contamination for radiocarbon analysis of dissolved organic carbon (Druffel et al., 2016; Druffel et al., 2019). Nevertheless, as filtration is a standard practice in oceanographic research, it can still be employed where necessary, particularly to reduce the potential risk of contamination from particulate matter and ensure sample homogeneity. In the coastal ocean, however, a notable difference was observed in the T peak

concentrations between unfiltered and filtered samples (0.7 and 0.2 µm), whereas the C and M peaks remained consistent. The high concentrations of T peak in unfiltered samples could be due to a fresh protein-like organic component, which has relatively large particle sizes (Lin and Guo, 2020). Therefore, while filtration appears to have minimal impact on C and M peaks in the open and coastal oceans, careful consideration of size fractionation is recommended for T peak measurements, especially for coastal water samples.

For the sample storage, C and M peaks in the open- and coastal-ocean waters can be preserved up to 21 days when stored in a refrigerator and freezer, regardless of whether samples were filtered or not (Fig. 4). However, at room temperature, significant changes in the concentrations of the C and M peaks were observed after five days, particularly for the M peak in the open ocean and the both C and M peaks in the coastal ocean (Fig. 5). Thus, storage of C and M peaks samples at room temperature for more than five days is not recommended in any sampling conditions. Unlike C and M peaks, the T peak showed significant changes within five days for any type of storage and filtration. These changes in T peak were presumably associated with rapid production and/or biodegradation of protein-like DOM. Thus, immediate measurements are required to accurately measure the T peak, which is biologically labile.

## 5. Conclusions

We investigated the effects of filter blanks, filter pore sizes, and storage strategies for measuring FDOM using seawater samples from the open and coastal oceans. We observed high blank values of FDOM originating from the filter without pre-washing and ashing procedures. The concentrations of C and M peaks were not affected by filter pore sizes for open- and coastal-ocean samples. However, filter pore sizes affected the T peak concentrations significantly, showing higher concentrations from 48% to 79% (unfiltered) in the coastal ocean samples than in the filtered samples. These findings suggested a possible option for measuring C and M peaks without filtration in the open ocean. Nonetheless, filtration is recommended to ensure consistent and accurate measurements. The concentrations of C and M peaks in seawater samples can be preserved for up to 21 days in a refrigerator or freezer, regardless of whether the samples are filtered or unfiltered. However, the concentration of T peak, even in filtered samples, rapidly decreased within five days, regardless of storage temperature. Overall,

if only the C and M peaks data are required, samples can be stored in the refrigerator and measured within 21 days. However, for T peak measurements, filtered samples should be immediately measured after sampling to prohibit misinterpretation.

**Data availability**

The original contributions presented in the study are included in the Supplementary Material; further inquiries can be directed to the corresponding author/s.

**Author Contribution Statements**

JS and GK contributed to the conceptualization of the study. JS and HH performed sampling, experiments, and analyses. JS,
IK, and GK were involved in the data interpretation and writing of the manuscript.

**Competing interests**

The authors declare that they have no conflict of interest.

**Acknowledgements**

We thank the members of the Environmental and Marine Biogeochemistry Laboratory (EMBL), SNU, for their assistance with
sampling and lab analysis. This study was funded by the Korea Institute of Ocean Science and Technology Research grant (KIOST, PEA0174). This study was also supported by the Korea Institute of Marine Science and Technology Promotion (KIMST), funded by the Ministry of Ocean and Fisheries, Korea (RS-2022-KS221662).

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
