# Peer review of "Technical note: Measurements of fluorescent dissolved organic matter (FDOM) in seawater (Filter blanks, pore sizes, and storage)"

_EGUsphere, 2025_

## Referee Comment (RC1)

**Review manuscript egusphere-2025-501**

**General comments**

This study examines the influence of filter blanks, pore sizes, and storage conditions on fluorescent dissolved organic matter (FDOM) measurements in seawater. Given the importance of FDOM as a tracer in oceanographic research, ensuring the accuracy and consistency of its measurements across different sampling protocols is critical. The manuscript presents clear objectives and a thorough discussion of the findings.

The study's primary objective is to refine methodologies for more reliable FDOM measurements. To achieve this, the researchers evaluated uncertainties associated with various sample preparation techniques. Using parallel factor analysis (PARAFAC), they identified three key FDOM components: the terrestrial humic-like peak (C peak), the marine humic-like peak (M peak), and the protein-like peak (T peak).

The main findings of the study are as follows:

1. **Effect of blanks**: Systematic blank values were observed when samples were filtered through pre-combusted glass fiber filters (0.7 μm pore size) and membrane filters (0.2 μm pore size) without pre-cleaning. This indicates that the filtration process can introduce contamination that affects FDOM measurements.

2. **Effect of pre-washing**: The blank values became negligible when the filters were pre-washed with 5 mL of 0.1 M HCl or 20 mL of distilled water, suggesting that proper pre-treatment of filters is essential for accurate measurements.

3. **Effect of storage**: $FDOM^H$ concentrations remained consistent for 21 days when stored in a refrigerator or freezer. However, the T peak concentrations showed significant decreases within five days, indicating that protein-like components are more susceptible to degradation.

I recognize the effort put into this study and the importance of the findings. However, I disagree with one recommendation made in the manuscript that can potentially have important impacts. In the discussion, the authors state:

> *the filtration procedure is not necessary for measuring FDOM concentrations in open ocean waters*

This statement seems counterintuitive, as filtration is a standard practice in oceanographic research to remove particulate matter and ensure sample homogeneity.

- First, you never know what you might find in a sample and it might be too late when you figure out that you should have filtered it.
- Second, based on Figure 3, filtering appears to have a significant effect on the T component. From the data provided in the appendix, I extracted the T peak concentrations for different storage conditions and measurement days (0, 5, and 21 days). The boxplot below illustrates that the T peak concentrations for unfiltered samples can be significantly higher than those for filtered samples. Therefore, I would advise caution in recommending against filtering samples.

[Figure]

Figure 1: Boxplot of T peak concentrations for different storage conditions and measurement days.

I also wonder if there are many researchers that do not filter their samples. I would suggest discussing this point further and providing more context on the implications of not filtering samples in oceanographic research.

**Specific comments**

**Abstract**

- Line 13: *the samples identified three..*: the samples cannot identify, the researchers or the statistics identified.

- Line 14: what is a *high procedural blank*? This should be rephrased for clarity. Maybe something like *high blank values*.

- Line 18: Define the H in FDOM$^H$. I guess this means humic-like, but it should be spelled out the first time.

- Line 21: It is said that significant bacterial degradation occurs after five days, but later in the same sentence it is said three days. Please clarify.

**Introduction**
- Line 33: I suggest removing the last sentence.

**Methods**
- Authors used 1 cm quartz cuvettes for the measurements. It seems a bit unusual to use so small pathlength cuvettes for samples from the open ocean. Could you provide more details on this choice?

- Line 59: It is specified that the deepest samples were measured at 500 meters, but in Fig. 4, we can see 2300 meters. Please clarify.

- Line 86: Maybe rephrase the sentence to be precise that the components were compared to the components found in the OpenFluor database.

- Line 95: The term *the filter blank* is often used. I would suggest replacing it with something like *the fluorescence of the C/M/T peaks measured on the filtrate*, or something similar.

**Results**
- The authors report some statistical results in the text, but they do not specify which tests were used.

- Figure 3:

  ‣ Why is there a second y-axis on the right side of the plot? It is confusing and unnecessary. Both unfiltered and filtered samples should be on the same y-axis (they already are on the same scale).
  ‣ Add the legend for the circle and triangle markers in the legend.
  ‣ Add a tag/annotation on each subplot to identify them in the text.

**Discussion**
- Line 133: I suggest replacing *in the course of water filtration* with *by the filtration process*.

- Line 157: I think you meant Fig. 4.

- Line 158: The authors say *However, at room temperature, we observed significant changes in FDOMH concentrations after five days*. However, I do not see many changes in the figure for peaks C and M. Do you mean the T peak?

**Conclusions**

In the conclusions and elsewhere, the authors suggest that the decision to filter or not should depend on the research question—specifically, whether the T peak is of interest. Could the authors clarify what types of research questions would not require consideration of the T peak? Providing a few examples would be valuable.

**Other comments**

- The samples come from different areas with varying salinity. Can this influence fluorescence intensity?

- Please identify each panel in the figures and cite them properly in the text when discussing them.

---

## Referee Comment (RC2)

[referee-annotated manuscript omitted]

---

## Author Comment (AC2)

**Review manuscript egusphere-2025-501**

**General comments**

Seo et al. present a very useful study on how the preprocessing, storage and filtration of seawater samples for FDOM measurements affects the results. This work has great value to the community, and somewhat suprising such studies have not been (to my knowledge) conducted to this degree. It's well written and concise. I recommned this is published with minor revision, which are largely to improve the clarity of presentations and provide the reader with some more useful details, as per below.

➔ Thank you for your valuable comments. In the revised manuscript, we carefully incorporated the comments below to improve clarity.

In the attached annotated ms. provided several comments and suggestions, which largely would improve the clarity of presentation. For the reader its much beneficial if some more details are given on several aspects of the methods used. Some key points include;

Provide the reader with the exact make and type of membrane filter used. Please also make sure the diameter of the filters is noted, since the volumes used for washing might be depedent on the filter size (surface area), and thus the reader needs to know how the used washing volumes relate to filter surface area.

➔ Additional information related to the membrane filter has been added to the revised manuscript.

Provide clearly (in a table), the initial concentrations of CDOM and FDOM in the filtered and unfiltered samples. This heklps the reader judge the type of samples that have been used.

➔ The initial concentrations have been provided in a table for clarification.

It would be also valuble to show whether CDOM changed over the same period of time, as this could affect the inner filter correction.

➔ The absorbance data from the samples will be provided in the revised version.

Any ancillary data that could tell about particle loading (e.g. CHLA, POC, etc.) in the samples used in the study would be also very valuable for the reader to evaluate how particle loading might ahve affected the results obtained.

➔ Unfortunately, no ancillary data were available for the samples in this study.

When in the field, sometimes neither distilled water nor HCL is available or practical to use, then the procedure would be to use sample water to pre-rinse the filter. Would this not be as efficient as using either of the above?

➔ Pre-rinsing the filter with the sample water can be considered as one of the available methods. This recommendation has been added to the revised version.

Several technical comments/suggestions are in the attached annotated ms.

➔ The comments have been carefully addressed and incorporated into the revised manuscript.

---

## Author Comment (AC4)

**Associate Editor's comments**

Line 31-32: Siegel et al. (2002) studied CDM (colored dissolved and detrital organic materials) not FDOM (FDOM is only part of CDOM). In addition, it is the absorption of UV by CDOM not FDOM that affects photosynthesis and the growth of marine microorganisms. Do not mess up FDOM with CDOM. These references are not appropriate for the statements that you made for FDOM.

➜ As pointed out by reviewer 2, this paragraph has been revised to more closely realtaed to FDOM.

Line 42: "formalin". I do not think Spencer et al. (2007) used formalin. Please check. If you find it in their paper, please let me know on what page it appears.

➜ Thank you for pointing this out. We acknowledge this citation was incorrect and have removed it from revised manuscript.

Line 46-47: "various sampling and storage methods". The preceding two statements appear to indicate that there is a generally accepted protocol for FDOM sampling and sample treatment. Reword the preceding statements.

➜ We have revised this statement to clarify that there is currently no standardized protocol with careful evaluation, and that various approaches are used depending on the research context.

Line 47: "for different DOM compositions". You were focused on FDOM earlier now suddenly switched to DOM. Be consistent.

➜ The original phrase "for different DOM compositions" was intended to refer to various environmental contexts. To clarify this point, we have revised the wording to "marine environment" in the revised manuscript.

Line 86: Coble (1996) and Coble et al. (1998) did not report PARAFAC modeling.

➜ This sentence has been revised as follow:

"Based on reference EEMs (Coble 1996; Coble et al., 1998; Coble, 2007), the terrestrial humic-like peak (C peak, Ex/Em = 375/457 nm), the marine humic-like peak (M peak, Ex/Em = 315/391 nm), and the protein-like peak (T peak, Ex/Em = 270/313 nm) were identified by the PARAFAC model.

Section 3.2: Add a comparison between the 0.2 um and 0.7 um filtration?

➜ A comparison between the 0.7 µm and 0.2 µm pore sizes has been added as follows:

"In both open-ocean and coastal-ocean samples, no significant difference in FDOM concentrations was observed between the 0.7 µm and 0.2 µm filtration."

Section 3.3: Add comparisons among different depths and different filter pore sizes? Figures show that different depths and pore sizes gave different results.

➜ For filter pore sizes, more explanations as described above has been added in Section 3.2. A comparison between the different depths has been added as follows:

"The concentrations of the C and T peaks in the open ocean showed no clear differences across

depths, whereas the concentration of the M peak was relatively lower at 2300 m compared to the other depths."

---

## Author Response (AR1)

**Associate Editor's comments**

1. Both reviewers disagreed with your recommendation that open-ocean waters do not need to be filtered for FDOM measurement (Reviewer#2's comment on this point is in the annotated manuscript copy)). Reviewer#1 even used your own data to support his/her argument. This is an important comment but your response, "We sincerely appreciate your insightful comments and will carefully incorporate them into the revised manuscript. Regarding the filtration process for FDOM measurements, we will tone down the description to account for the potential influence of contamination", essentially does not contain any useful information with respect to the reviewers' comments. Please directly address the reviewers' concern in sufficient detail.

➜ We have revised the discussion as follow:

(line 173-185) "The concentrations of all FDOM components in the open ocean showed no significant differences between unfiltered and filtered samples (0.7 and 0.2 µm). This result suggests that the presence of particles in the sample did not significantly affect the fluorescent concentration in the open ocean. These results indicate that filtration may not be essential for measuring C and M peaks in open ocean waters, and measurements can be reliably conducted without filtration. Similarly, in the open-ocean samples, filtration was not conducted to avoid potential contamination for radiocarbon analysis of particulate organic carbon (Druffel et al., 2016; Druffel et al., 2019). Nevertheless, as filtration is a standard practice in oceanographic research, it can still be employed where necessary, particularly to reduce the potential risk of contamination from particulate matter and ensure sample homogeneity. In the coastal ocean, on the other hands, a notable difference was observed in the T peak concentrations between unfiltered and filtered samples (0.7 and 0.2 µm), whereas the C and M peaks remained consistent. The high concentrations of T peak in unfiltered samples could be due to fresh protein-like organic component, which has relatively large particle sizes (Lin and Guo, 2020). Therefore, while filtration appears to have minimal impact on C and M peaks in the open- and coastal- ocean, careful consideration of size fractionation is recommended for T peak measurements, especially for coastal water samples."

2. Comment by Reviewer#2: "line 30-33. Note, it rather the CDOM part of DOM that affects the light, please be clear about that, FDOM as such, with fluorescence would have a lesser role in affecting the light.. a prerequisite for FDOM is that CDOM absorb light first ...". Your response, "This part has been revised to be more closely related to FDOM", again does not provide the information on what specific revision was made to address the reviewer's comment.
➔ The introduction has been revised as follow:
(line 26-36) "Fluorescent dissolved organic matter (FDOM), which emits fluorescent light after absorbing energy, is ubiquitous in the ocean and provides important information on the origins and behavior of dissolved organic matter (DOM) in the ocean (Nelson and Siegel, 2013; Stedmon and Nelson, 2015). FDOM in ocean waters is generally classified into two groups (humic-like and protein-like substances) based on the excitation and emission spectrum (Coble, 2007). The humic-like component is primarily derived from microbial decomposition of organic debris in sediments and soils, as well as from materials sinking through the water column as marine snow (Yamashita and Tanoue, 2008; Yamashita and Tanoue, 2009). In contrast, the protein-like component represents a more labile fraction of DOM and is mainly produced through primary production and biological activity in the surface waters (Lønborg et al., 2010). Based on distinct fluorescent properties, FDOM has been used as a tracer for water circulation in the ocean (Galletti et al., 2019; Margolin et al., 2018), estimating DOM turnover times in the global ocean (Catalá et al., 2015), and for calculating the fractions of different water masses in the ocean (Kim et al., 2020; Wang et al., 2022)."

3. Comment by Reivewer#1: The authors report some statistical results in the text, but they do not specify which tests were used". Your response is "In the revised version, we have added a brief explanation in this section where the p-value is first introduced (line 118), clarifying that p values higher than 0.05 are considered not statistically significant." Please clearly state what statistical test(s) were conducted. Note that this issue had been raised by me twice during the initial review but was not been appropriately addressed.
➔ A t-test was conducted, and this information has been added to the Results section of the revised manuscript.

4. In addition to the separate comments posted in the reviewer's report, Reveiwer#2 also thoroughly annotated the manuscript on both the scientific and technical aspects. I requested earlier that you address the comments in the annotated copy as well but it seems that you chose to ignore them. Here, I request again that you provide point-by-point responses to the comments in the annotated manuscript (you can copy and paste the reviewer's comments to facilitate your response).
➔ The responses for the comments in the annotated manuscript have been included in the

Line 31-32: Siegel et al. (2002) studied CDM (colored dissolved and detrital organic materials) not FDOM (FDOM is only part of CDOM). In addition, it is the absorption of UV by CDOM not FDOM that affects photosynthesis and the growth of marine microorganisms. Do not mess up FDOM with CDOM. These references are not appropriate for the statements that you made for FDOM.
➔ As pointed out by reviewer 2, this paragraph has been revised to more closely related to FDOM.

Line 42: "formalin". I do not think Spencer et al. (2007) used formalin. Please check. If you find it in their paper, please let me know on what page it appears.
➔ Thank you for pointing this out. We acknowledge this citation was incorrect and added other reference (Wurl et al., 2009).

Line 46-47: "various sampling and storage methods". The preceding two statements appear to indicate that there is a generally accepted protocol for FDOM sampling and sample treatment. Reword the preceding statements.
➔ We have revised this statement to clarify that there is currently no standardized protocol with careful evaluation, and that various approaches are used depending on the research context.

Line 47: "for different DOM compositions". You were focused on FDOM earlier now suddenly switched to DOM. Be consistent.
➔ The original phrase "for different DOM compositions" was intended to refer to various environmental contexts. To clarify this point, we have revised the wording to "marine environment" in the revised manuscript.

Line 86: Coble (1996) and Coble et al. (1998) did not report PARAFAC modeling.
➔ This sentence has been revised as follow:
(lines 89-92) "Terrestrial humic-like peak (C peak, Ex/Em = 375/457 nm), marine humic-like peak (M peak, Ex/Em = 315/391 nm), and protein-like peak (T peak, Ex/Em = 270/313 nm) were identified by the PARAFAC model. These peak positions were assigned based on the previously reported values (Coble, 1996; Coble et al., 1998; Coble, 2007)."

Section 3.2: Add a comparison between the 0.2 um and 0.7 um filtration?
➔ A comparison between the 0.7 µm and 0.2 µm pore sizes has been added as follows:
(lines 133-134) "In both open-ocean and coastal-ocean samples, no significant difference in FDOM concentrations was observed between the 0.7 µm and 0.2 µm filtration."

Section 3.3: Add comparisons among different depths and different filter pore sizes? Figures show that different depths and pore sizes gave different results.
➔ For filter pore sizes, more explanations as described above has been added in Section 3.2. A comparison between the different depths has been added as follows:
(lines 117-118) "The concentrations of the C and T peaks in the open ocean showed no clear

differences across depths, whereas the concentration of the M peak was relatively higher at 300 m compared to the other depths."

**Reviewer #1's comments**
**General comments**

This study examines the influence of filter blanks, pore sizes, and storage conditions on fluorescent dissolved organic matter (FDOM) measurements in seawater. Given the importance of FDOM as a tracer in oceanographic research, ensuring the accuracy and consistency of its measurements across different sampling protocols is critical. The manuscript presents clear objectives and a thorough discussion of the findings.

The study's primary objective is to refine methodologies for more reliable FDOM measurements. To achieve this, the researchers evaluated uncertainties associated with various sample preparation techniques. Using parallel factor analysis (PARAFAC), they identified three key FDOM components: the terrestrial humic-like peak (C peak), the marine humic-like peak (M peak), and the protein-like peak (T peak).

The main findings of the study are as follows:

1. Effect of blanks: Systematic blank values were observed when samples were filtered through pre-combusted glass fiber filters (0.7 μm pore size) and membrane filters (0.2 μm pore size) without pre-cleaning. This indicates that the filtration process can introduce contamination that affects FDOM measurements.

2. Effect of pre-washing: The blank values became negligible when the filters were pre-washed with 5 mL of 0.1 M HCl or 20 mL of distilled water, suggesting that proper pre-treatment of filters is essential for accurate measurements.

3. Effect of storage: $FDOM_H$ concentrations remained consistent for 21 days when stored in a refrigerator or freezer. However, the T peak concentrations showed significant decreases within five days, indicating that protein-like components are more susceptible to degradation.

I recognize the effort put into this study and the importance of the findings. However, I disagree with one recommendation made in the manuscript that can potentially have important impacts. In the discussion, the authors state:

*the filtration procedure is not necessary for measuring FDOM concentrations in open ocean waters*

This statement seems counterintuitive, as filtration is a standard practice in oceanographic research to remove particulate matter and ensure sample homogeneity.

• First, you never know what you might find in a sample and it might be too late when you figure out that you should have filtered it.

• Second, based on Figure 3, filtering appears to have a significant effect on the T component. From the data provided in the appendix, I extracted the T peak concentrations for different storage conditions and measurement days (0, 5, and 21 days). The boxplot below illustrates

that the T peak concentrations for unfiltered samples can be significantly higher than those for filtered samples. Therefore, I would advise caution in recommending against filtering samples.

[Figure]

Figure 1 Boxplot of T peak concentrations for different storage conditions and measurement days.

I also wonder if there are many researchers that do not filter their samples. I would suggest discussing this point further and providing more context on the implications of not filtering samples in oceanographic research.

➜ We sincerely appreciate your insightful comments and will carefully incorporate them into the revised manuscript. Regarding the filtration process for FDOM measurements, we will tone down the description to account for the potential influence of contamination.

**Specific comments**

**Abstract**

• Line 13: *the samples identified three..*: the samples cannot identify, the researchers or the statistics identified.
➜ This sentence has been rephrased as follow:
   (lines 13-15) "Three primary components were identified from these samples using parallel factor analysis: terrestrial humic-like peak (C peak), marine humic-like peak (M peak), and protein-like peak (T peak)."

• Line 14: what is a *high procedural blank*? This should be rephrased for clarity. Maybe something like *high blank* values.

➔ This sentence has been revised to clarify the meaning, as follow:
  (lines 15) "Relatively high blank values were observed …"

• Line 18: Define the H in FDOM_H. I guess this means humic-like, but it should be spelled out the first time.
➔ All abbreviations are defined upon their first time in the manuscript.

• Line 21: It is said that significant bacterial degradation occurs after five days, but later in the same sentence it is said three days. Please clarify.
➔ removed for clarity

**Introduction**

• Line 33: I suggest removing the last sentence.
➔ removed

**Methods**

• Authors used 1 cm quartz cuvettes for the measurements. It seems a bit unusual to use so small pathlength cuvettes for samples from the open ocean. Could you provide more details on this choice?
➔ The path length of quartz cuvettes depended on the FDOM concentrations. (lines 75-76) In marine environments, a 1 cm path length quartz cuvette was used, which is a standard and widely adopted practice in FDOM measurements (Murphy et al., 2013).

Reference:
Murphy, K. R., Stedmon, C. A., Graeber, D., & Bro, R. (2013). Fluorescence spectroscopy and multi-way techniques. PARAFAC. Analytical methods, 5(23), 6557-6566."

• Line 59: It is specified that the deepest samples were measured at 500 meters, but in Fig. 4, we can see 2300 meters. Please clarify.
➔ (line 62) Sampling was conducted at surface (3 m), 300 m, 2300 m, and this correction has been incorporated into the revised manuscript.

• Line 86: Maybe rephrase the sentence to be precise that the components were compared to the components found in the OpenFluor database.
➔ revised to clarify as follow:
   (lines 93-95) "The fluorescence spectra results were compared with the OpenFluor spectra database (Murphy et al., 2014), and statistical matches were found at a confidence level of > 95%, with 28 matches for the C peak, 37 for the M peak, and 45 for the T peak."

• Line 95: The term the filter blank is often used. I would suggest replacing it with something like the fluorescence of the C/M/T peaks measured on the filtrate, or something similar.

➔ The term "filter blank" was changed to "fluorescence of the C/M/T peaks measured on the filtrate" in the revised manuscript.

**Results**

• The authors report some statistical results in the text, but they do not specify which tests were used.

➔ In the revised version, we conducted t-test and have added a brief explanation in this section where the p-value is first introduced (lines 128-129), clarifying that p values higher than 0.05 are considered not statistically significant.

• Figure 3:
‣ Why is there a second y-axis on the right side of the plot? It is confusing and unnecessary. Both unfiltered and filtered samples should be on the same y-axis (they already are on the same scale).
‣ Add the legend for the circle and triangle markers in the legend.
‣ Add a tag/annotation on each subplot to identify them in the text.

➔ The y-axis corresponding to the unfiltered samples has been removed, and the data are presented together with the filtered samples, as per reviewer's suggestion. Annotations from (a) to (c) will be added to distinguish each subplot, and a description for each subplot will be provided in the figure caption.

**Discussion**

• Line 133: I suggest replacing *in the course of water filtration with by the filtration process*.

➔ (lines 159-160) The phrase "in the course of water filtration and sample storage" was changed to "in the course of water filtration with by the filtration process".

• Line 157: I think you meant Fig. 4.

➔ corrected

• Line 158: The authors say *However*, *at room temperature*, we observed *significant changes in FDOM$_H$ concentrations after five days*. However, I do not see many changes in the figure for peaks C and M. Do you mean the T peak?

➔ FDOM$_H$ concentrations after five days showed significantly differences compared to those within five days, particularly for the M peak in the open ocean and the C and M peaks in the coastal ocean (Fig. 5). In the revised manuscript (lines 189-190), we have incorporated this

information to improve the clarity.

**Conclusions**

In the conclusions and elsewhere, the authors suggest that the decision to filter or not should depend on the research question—specifically, whether the T peak is of interest. Could the authors clarify what types of research questions would not require consideration of the T peak? Providing a few examples would be valuable.

➔ As suggested, we have clarified in the revised manuscript the types of research questions that may not require consideration of the T peak.

 For example:

1) In open ocean observations where sample storages typically exceeds one weeks before analysis, degradation of the T peak over time can lead to misinterpretation.

2) In costal studies focused on identifying the sources of humic substances, rather than investigating biological processes.

**Other comments**

• The samples come from different areas with varying salinity. Can this influence fluorescence intensity?

➔ In general, FDOM components show various intensity range depending on different sampling stations (these can be either coastal vs. open ocean, surface layer vs. benthic condition), but the salinity effect is not clear. For example, in the open ocean, the T peak exhibited high intensity in the biologically productive surface layer and decreased sharply with depth. In addition, $FDOM_H$ showed low intensity at the surface and gradually increased with depth.

• Please identify each panel in the figures and cite them properly in the text when discussing them.

➔ corrected

**Reviewer #2's comments**
**General comments**

Seo et al. present a very useful study on how the preprocessing, storage and filtration of seawater samples for FDOM measurements affects the results. This work has great value to the community, and somewhat suprising such studies have not been (to my knowledge) conducted to this degree. It's well written and concise. I recommned this is published with minor revision, which are largely to improve the clarity of presentations and provide the reader with some more useful details, as per below.

➔ Thank you for your valuable comments. In the revised manuscript, we carefully incorporated the comments below to improve clarity.

In the attached annotated ms. provided several comments and suggestions, which largely would improve the clarity of presentation. For the reader its much beneficial if some more details are given on several aspects of the methods used. Some key points include;

Provide the reader with the exact make and type of membrane filter used. Please also make sure the diameter of the filters is noted, since the volumes used for washing might be depedent on the filter size (surface area), and thus the reader needs to know how the used washing volumes relate to filter surface area.

➔ (line 54 and 56) Additional information, including the materials, pore-sizes, and diameters of the GF/F and membrane filters, has been added in Section 2.1 of the Methods.

Provide clearly (in a table), the initial concentrations of CDOM and FDOM in the filtered and unfiltered samples. This helps the reader judge the type of samples that have been used.

➔ (lines 115-125) The initial concentrations are presented in Section 3.2 of the Results, and will also be provided in a table for clarity.

It would be also valuable to show whether CDOM changed over the same period of time, as this could affect the inner filter correction.

➔ We agree that CDOM is beyond the scope of the current study. This research specifically focused on the characterization of FDOM; therefore, CDOM was not included in the experimental design.

Any ancillary data that could tell about particle loading (e.g. CHLA, POC, etc.) in the samples used in the study would be also very valuable for the reader to evaluate how particle loading might ahve affected the results obtained.

➔ Unfortunately, no ancillary data were available for the samples used in this study.

When in the field, sometimes neither distilled water nor HCL is available or practical to use, then the procedure would be to use sample water to pre-rinse the filter. Would this not be as

efficient as using either of the above?

➔ (lines 172-173) Pre-rinsing the filter with sample can be considered one of the available methods to reduce contamination. This recommendation has been added to the Discussion section in the revised manuscript.

**[Technical comments/suggestions in annotated manuscript]**

line 11. DOM needs to be written out first time its used..

➔ (line 11) In the revised manuscript, "DOM" has been spelled out as "dissolved organic matter".

line 11. Re-word; "results from FDOM measurement can depend on ..."

➔ (line 11) changed as suggested

line 16. specify which type of membrane was used .. so many different membranes out there..

➔ (lines 15-16) In the revised manuscript, we have added detailed information on the materials and diameters of both the GF/F and membrane filters.

line 16. please clarify, was this used for both type of filter..

➔ (line 17) Correct, we have included information regarding both filter.

line 18-19. does this suggest that T peak material is primarily associated with particulates, and samples need to be filtered to not get material from broken cells into the dissolved phase?

➔ (lines 22-23) The T peak is associated with biologically derived materials and reflects the presence of relatively larger particles produced by biological activity, in contrast to the smaller-sized humic substances represented by the C and M peaks. Therefore, this finding suggests that consistent filtration through the same pore size is necessary when comparing the relative concentrations of the T peak.

line 20. in the dark?

➔ (line 20) An amber vial was used for sample storage, and this information has been added.

line 22. applies for C and M peak only, please clearly state that at the start of the sentence

➔ Throughout the manuscript, all occurrences of "$FDOM_H$" have been spelled out as "C and M peaks".

line 22. I would disagree, does not the above statements mean that material in the T peak could in seawater be often associated with particles, and thus using an unfiltered samples could caus misleading results for this peak?

➔ (lines 20-21) This section specifically addresses the C and M peaks. To improve clarity, we have revised the text to separately describe the findings related to the C and M peaks and those for the T peak.

line 30-33. Note, it rather the CDOM part of DOM that affects the light, please be clear about that, FDOM as such, with fluorescence would have a lesser role in affecting the light.. a prerequisite for FDOM is that CDOM absorb light first ...

➔ (lines 28-36) This part has been revised to be more closely related to FDOM.

line 52. Please specify which exact type of membrane filter was used, this is a great interest to

the reader. Simply broadly stating the material, does not mean other similar membranes from other manufacturers are exactly the same. Also note what diameter the filter used was. This is related to volume needed for washing, as the larger the surface area, the more volume would be needed..

➔ (line 54 and 56) Filter information including material, manufacture, pore-size, and diameter has been added in the revised manuscript.

line 62-63. stored how, and for how long before filtration?

➔ (line 65) Samples were stored in refrigerator. The filtration of samples was conducted within two days after seawater sampling. To clarity, this information added in the revised manuscript.

line 64. Please specify the equipment used to do the filtration, was this on a funnel system, syringe filter, or? Please also note how this equipment was cleaned and how carry over from previous sample was avoided? And I assume samples were stored in the dark, despite being in amber vials?

➔ (lines 67-68) Filtration was performed using a 47 mm diameter funnel system that had been cleaned with 1 M HCl. Between samples, the funnel and filter system were rinsed with distilled water and next samples. Although all samples were stored in the dark container, photodegradation may have occurred during equilibration to room temperature prior to measurement.

line 98. For the reader it would be really good if a Table was prepared where the initial concentration of CDOM and FDOM for the samples used in this study are listed. This would indicate the range of CDOM and FDOM values used.

➔ (lines 115-125) The initial concentrations will be presented in the Results section and also provided in a table for clarity.

Figure 4. Also show this for CDOM, or at least make a note how CDOM changed.

➔ We acknowledge the potential importance of the relationship between CDOM and FDOM. However, this study did not incorporate an experimental design specifically targeting CDOM, and as such, we are limited in our ability to provide CDOM data.

Figure 5. add crosses to the legend as in the previous figure

➔ The legend has been added in the revised manuscript.

line 137. This volume must be dependent on filter surface area, so please, make sure to also disclose this information clearly, and relate to the filter surface area (i.e. volume per filter surface area)..

➔Information on the filter diameter and the volume per filter surface area (lines 102-103) is provided in the Methods and Results sections.

line 139. not just activity, but that cells pass the filter, and material from these are measured as "dissolved"..?

➔ (lines 164-165) Correct. Filtration using GF/F can lead to misleading FDOM concentration due to the passage of microorganisms and cells. However, most research group that measure FDOM with DOC to trace the source of DOM have used 0.7 um pore-size filter (GF/F), owing to their advantages such as low background DOM, high flow rate, and large capacity.

Additional explanation on this point has been included in the revised manuscript.

line 140. again, I would rather phrase this not only as activity,. but also such that particles might pass the filters, and then either by abiotic processes (cell bursting?) or microbial activity be producing measurable FDOM

➔ (lines 164-165) As described above, this comment has been incorporated into the revised manuscript.

line 141. please remind the reader of the exact membrane material

➔ The material of membrane filter (mixed cellulose ester) has been provided in the revised manuscript.

line 147. Do you have any measure of e.g. CHLA or POC in these samples, to how different the samples are in terms of particle loading, please given any such data, tabulated together with CDOM and CDOM data.

➔ Unfortunately, no measurements of other parameters were conducted in this study.

line 148-149. This is a bold statement, since if there are high phytoplankton biomass, at least the T peak could be affected? Please clarify the statement, I think this is only valid, if there is low particle loading.

➔ (lines 174-186) This statement has been revised to focus on humic-like FDOM peaks. As pointed out by Reviewer 1, the discussion regarding sample filtration has also been toned down accordingly.

line 152. please simply spell out the peaks here in the text.. e.g. "humic-like C and M peaks.

➔ revised as suggested

line 153. spell out the peaks here, simply more clearer for the reader. if you always spell out the peaks..

➔ revised as suggested

line 163. Conclusion - I would have expected one clear sentence on the recommendation for how filters are cleaned..

➔ The recommendation for the sample filtration has been added as below:
  (lines 200-201) "These findings suggested a possible option for measuring FDOM without filtration in the open ocean."

line 164. also filter type?

➔ Although GF/F and membrane filters were used in this study to investigate filter blanks and pore sizes, we do not intend to recommend the exclusive use of these two filter types for FDOM analysis. Therefore, we have chosen not to specify or emphasize the filter type in the context of general recommendations.

line 169. spell out the peaks.. or refer to "humic-like C and M peaks" ...

➔ revised as suggested

line 171. spell out, "humic-like"

➔ revised as suggested

---

## Author Response (AR2)

**Associate Editor's comments**

➔ Thanks for the valuable comments from the associate editor and reviewer #2. In this stage, we carefully revised our manuscript following the below comments.

L44-45. Spencer et al. (2007) did not use formalin in their study. Wurl et al. (2009) did use formalin as a sample preservative but they did not measure FDOM. Instead, they measured CDOM absorption coefficients but did not report if formalin had any artifacts on their measurements. Please fully digest the references before considering citing them.

➔ removed as suggested

**Reviewer #2's comments**

Line 21. I would perhaps explicititly state also that ", room temperature storage is not recommended."

➔ We added more explanations as follows:

"In contrast, clear changes were observed in samples stored at room temperature after five days."

Line 34. There are earlier papers that use FDOM to traer water masses, and I would suggest the authors explore that, to do justice of earlier work. To my knowledge at least this paper did it in the Arctic: https://www.nature.com/articles/srep33978 (could be added here).

➔ added as suggested

"Gonçalves-Araujo, R., Granskog, M. A., Bracher, A., Azetsu-Scott, K., Dodd, P. A., and Stedmon, C. A.: Using fluorescent dissolved organic matter to trace and distinguish the origin of Arctic surface waters, Scientific Reports, 6, 33978, https://doi.org/10.1038/srep33978, 2016."

Line 37-48. This now reads like only GFF filters are used. However, different types of menbrane filters are also widely used. I think the authors should acknowledge this, since they also use a membrane filter in their study.

➔ We have added more explanation as follows:

"In some studies, different types of membrane filters, such as cellulose acetate, polyethersulfone, and polycarbonate, have also been used for FDOM measurement (Amaral et al., 2023; Chen et al., 2022; Vines and Terry, 2020)."

Table 1 - just a suggestion, easier to compare the different filtrations if you always have the different peaks side by side, e.g. first have C peak unfilters, C peak 0.7 filt., C peak 0.2 filt., etc. So sort columns by peak, not by unfiltered/filtration.

➔ Table 1 was revised as suggested.

Line 142-145 Perhaps note that samples were stored in the dark too in every case?

➜ In the Methods (line 70) and Results (line 145) sections, detailed information on sample storage is provided as follows:

"Unfiltered and filtered (0.7 or 0.2 μm) water samples (~ 40 mL each) were stored in the dark (i.e., in pre-combusted amber vials) to prevent photodegradation, and kept at three different temperatures (–20°C, 4°C, or 25°C)."

"The concentrations of C and M peaks for unfiltered and filtered samples (0.7 and 0.2 μm) from the open- and coastal-ocean stored in the refrigerator (4°C) and freezer (–20°C) under dark conditions, showed no clear differences between the initial and after 21 days measurements (8% ± 3%, p > 0.05) (Fig. 4)."

Line 159-160, with the addition of text, the sentence does not read properly, please chech and rephrase.

➜ This sentence has been rephrased as follows:

"The accuracy of FDOM measurements can be largely affected by the filtration process."

Line 164 and 169 - the volumes given are for a certain size of filter that is used here, so you need to give this volume rather by volume per filter area to indicate a more universal measure of the volume required.

➜ The volume per filter area information is also provided in the revised version.

"This result suggests that GF/F should be washed with 20 mL of distilled water (~1.16 mL cm$^{-2}$) or 5 mL of 0.1 M HCl (~0.29 mL cm$^{-2}$) before the sample filtration for FDOM measurements."

---

## Author Response (AR3)

**Associate Editor's comments**

Wurl et al. (2009) in the reference list is shown in the track-change mode. Clean it up.
➔ removed as suggested

Line 66: It's rare to call a 20-L container as a bottle. Change "bottles" to "carboys" or simply "containers".
➔ changed as suggested

Throughout the manuscript (including figures, tables): the units of fluorescence peaks (R.U.) are not concentration units. Avoid using "concentration or concentrations" to describe the magnitude of fluorescence peaks. Instead, use "fluorescence intensity (signal)" or "fluorescence intensities (signals)".
➔ In the whole manuscript, the word "concentration" was rephrased to "intensity".